# Peri- and Post-Menopausal Women with Schizophrenia and Related Disorders Are a Population with Specific Needs: A Narrative Review of Current Theories

**DOI:** 10.3390/jpm11090849

**Published:** 2021-08-27

**Authors:** Alexandre González-Rodríguez, Armand Guàrdia, José Antonio Monreal

**Affiliations:** 1Department of Mental Health, Mutua Terrassa University Hospital, University of Barcelona, 08221 Terrassa, Spain; aguardia@mutuaterrassa.cat (A.G.); jamonreal@mutuaterrassa.cat (J.A.M.); 2Institut de Neurociències, Universitat Autònoma de Barcelona (UAB), 08211 Terrassa, Spain; 3Centro de Investigación Biomédica en Red de Salud Mental (CIBERSAM), 08221 Terrassa, Spain

**Keywords:** schizophrenia, menopause, hormones, antipsychotics, comorbidity

## Abstract

Background: While gender differences in antipsychotic response have been recognized, the potential role of menopause in changing drug efficacy and clinical outcome in schizophrenia related disorders has been understudied. We aimed to review the relevant literature to test whether optimizing menopausal and post-menopausal treatment and addressing specific health needs of this stage in life will improve outcome. Methods: Non-systematic narrative review using the PubMed database (1900–July 2021) focusing on randomized controlled trial results addressing our question. Forty-nine studies met our criteria. Results: Premenopausal women show significantly better antipsychotic response than postmenopausal women. Hormone replacement therapies (HRT) should be used in postmenopausal women with schizophrenia with caution. Raloxifene, combined with antipsychotics, is effective for psychotic and cognitive symptoms in postmenopausal women with schizophrenia and related disorders. Medical comorbidities increase after menopause, but the influence of comorbidities on clinical outcomes has been poorly investigated. Preventive strategies include weighing risks and benefits of treatment, preventing medical comorbidities, and enhancing psychosocial support. Ideal treatment settings for this population warrant investigation. Conclusions: Antipsychotic dose adjustment at menopause is recommended for schizophrenia. Raloxifene may play an important role in permitting dose reduction and lessening adverse effects. Prevention of comorbidities will help to reduce the mortality rate.

## 1. Introduction

Aside from the all-important chromosomal difference between the sexes, levels of gonadal hormones are that which most distinguish males and females. As the differences in these levels markedly change at menopause, it is important to ask whether male/female differences observed in the expression and treatment of schizophrenia also change at menopause. For instance, during the reproductive years, women, on average, require significantly lower doses of antipsychotic medication (AP) than men to improve psychopathological symptoms [1]. Is this still true after menopause [2]? There is reason to believe that the loss of oestrogen takes this female advantage away [3]. Prior to menopause, cognitive defects are much more marked in men with schizophrenia when compared to women [4]. This female advantage also appears to wane with oestrogen decrease [5,6]. Moreover, preclinical studies support the theory that oestrogen loss increases vulnerability to psychosis [7]. For instance, using an animal model of psychosis (reversing disrupted latent inhibition), Arad and Weiner have shown that, in ovariectomized rats, co-administration of 17 beta-estradiol with haloperidol and clozapine increases the antipsychotic-like effect, particularly the effect on positive symptoms, as evaluated in animal models [8]. This suggests that the treatment of women with schizophrenia requires a post-reproductive stage focus that differs from that called for in standard treatment guidelines [9].

Several authors have described the relationship between estradiol levels and psychotic symptoms in other clinical conditions rather than schizophrenia [9]. These nosological entities have been called ‘the psychoses of the menstruation’ or menstrual psychoses, childbearing psychoses, or postpartum psychoses, and, in the case of psychoses at the time of menopause, the vast majority of authors have referred to them as postmenopausal psychoses [9]. All of these entities seem to share a common pathway indicating that the loss of oestrogen levels may eventually lead to an occurrence of psychotic symptoms or the risk of exacerbation in women with previous history of such conditions [2,3]. In the particular case of menstrual psychoses, theories have pointed out to infer a higher probability of the occurrence of psychotic symptoms in the low oestrogen levels phase [3]. At the postpartum period, estradiol levels drop from their peaks during pregnancy, suggesting that this decline can sometimes manifest as an appearance of delusions, hallucinations, or cognitive distortions. Perimenopausal and postmenopausal periods in a woman’s life are also times of special vulnerability as the permanent withdrawal of gonadal hormones is associated with mental distress [9]. In brief, a high amount of literature highlights the need to pay special attention to reproductive and post reproductive stages of women’s life [9].

It is possible that optimizing treatment for women after menopause will not change long term outcomes. What may be critical for women is the amount of oestrogen they have been exposed to prior to menopause. For instance, a study on the effect of life exposure to oestrogen on memory in women who have experienced natural menopause found that, the longer their reproductive period (menarche to menopause) had been, the better their postmenopausal cognition [10]. Hesson asked the same question and included more variables -e.g., duration of oestrogen therapy, age at menarche and menopause, body mass index at post menopause, time since menopause, nulliparity, and duration of breastfeeding [11]. This was a more accurate way of estimating exposure to oestrogens than duration of reproductive life alone. The results of this study echoed those of Tierney et al. [10]. An earlier study, however, had not been able to find an association [12].

Though no longitudinal studies exist, for the purpose of this review, we assume that optimizing menopausal and post-menopausal treatment of women with schizophrenia will improve their outcomes, both with respect to cognition and positive symptoms of psychosis.

Our review, therefore, asks the following questions: In postmenopausal women with schizophrenia and related disorders,

(1)Does the choice of antipsychotic (prolactin sparing vs. prolactin raising; weight gaining vs. weight neutral) make a difference?(2)Does raising the antipsychotic dose over premenopausal levels make a difference?(3)Does adding hormone replacement therapy make a difference?(4)Does adding therapeutic oestrogen or selective oestrogen receptor modulator make a difference?(5)Does attending to comorbidities make a difference?(6)Does family stress increase symptom level?(7)Does treatment setting and the comprehensiveness of care make a difference?

## 2. Methods

We carried out a non-systematic narrative review based on electronic searches using the PubMed database from 1900 until July 2021 and focusing on papers that addressed our questions. To start the search strategy in PubMed, we initially keyed in: (menopause OR post menopause OR postmenopausal) AND (schizophrenia OR psychosis). Google Scholar was also used to find relevant but non-medical papers unavailable on the PubMed database. Conference abstracts were excluded. Reference lists of retrieved studies were also searched, and cited papers were included if relevant. 

The screening and selection process was conducted by A.G.R. and A.G.D., who scanned abstracts and titles, and, on this basis, selected papers for further full-text scrutiny. We included several levels of evidence: Randomized Controlled Treatment (RTC) trial results alone or in reviews, results of cohort studies, alone or in reviews, epidemiological studies of outcomes, case control studies alone and in reviews, case series and expert critical appraisals based on epidemiology and/or basic research. Where papers referred to the same work, the earlier publication was the one included. We then critically appraised the selected papers. Ultimately, a total of 49 studies were identified as relevant to our seven questions. The flow chart of all included studies is described in Figure 1.

## 3. Results

From the initial 265 records, we used 49 studies as relevant to the questions we wanted to be able to answer. We then divided the review into the following sections: (1) choice of antipsychotic (prolactin sparing vs. prolactin raising) in postmenopausal women with schizophrenia and related disorders, (2) antipsychotic dose adjustment, (3) addition of hormone replacement therapy, (4) addition of therapeutic oestrogen or selective oestrogen modulators, (5) impact of menopausal symptoms and comorbidities on treatment decisions, (6) impact of family stress on symptom level, and (7) influence of treatment setting and comprehensiveness of care on outcomes in postmenopausal women with schizophrenia and related disorders.

The term neuroleptic refers to the extrapyramidal symptoms or neurological side effects associated with antipsychotic use. This is a category defined as per the neurological symptoms, and this term has persisted for many decades as the occurrence of extrapyramidal symptoms in patients receiving chlorpromazine was associated with the efficacy of this drug [13]. Several years later, the theory linking neurological side effects and drug efficacy was questioned. In fact, thioridazine was found to show similar efficacy compared to chlorpromazine, but the side effects were lower for the first drug. 

In a second step, the term ‘major tranquilizers’ has been preferred by several authors as it do not describe a category of drugs based on side-effects. It refers to a group of drugs by their efficacy or their role in improving clinical symptoms.

The term ‘antipsychotics’ has been later designed for the drugs acting on psychotic symptoms. The concept of atypical antipsychotics is a working definition taking special emphasis on their effects on dopaminergic and serotonergic pathways [14]. On the other hand, in the last decades, another term has emerged. Second-generation antipsychotics has been considered as antipsychotic medications opposed to the traditional or first-generation drugs. Olanzapine, risperidone, quetiapine, and clozapine are included in this group of second-generation antipsychotics, which has been considered for several authors insufficient or less comprehensive. The classification of aripiprazole, brexpiprazole, or cariprazine as third generation drugs has been less discussed. More recently, new terminological proposals such as serotonergic-dopaminergic antagonists have been used to explain the prediction of treatment response and the occurrence of side effects attributable to treatment.

### 3.1. Choice of Antipsychotics in Postmenopausal Women with Schizophrenia and Related Disorders

Usall and collaborators investigated gender differences in response to 1st and 2nd generation antipsychotics in a 3-year prospective observational study called the Schizophrenia Outpatient Health Outcomes study (SOHO), which recruited nearly 10,000 outpatients with schizophrenia from 10 European countries [15]. This is a large multicentre study including patients with schizophrenia which were divided into two groups according to the treatment they received: atypical antipsychotics or typical antipsychotics. Patients were treated with only one antipsychotic, as follows: olanzapine, risperidone, quetiapine, amisulpride, clozapine, oral typical antipsychotic, or depot typical antipsychotic. Other diagnoses rather than schizophrenia were not specified in the study. Patients were included if they were starting an antipsychotic for the first time or switching from a previous one. The investigators compared the use of olanzapine with other antipsychotics in two different cohorts of patients: olanzapine vs. non-olanzapine antipsychotics. Data were documented at baseline, after 3 and 6 months, and every 6 months up to 36 months after the initial treatment visit. Psychopathological symptoms were assessed using the Clinical Global Impression (CGI) scale for positive, negative, cognitive, depressive, and overall symptoms. Treatment tolerability included body mass index and registration of extrapyramidal side effects; quality of life was evaluated by means of the European Quality of Life-5 Dimensions (EQ-5D) scale. Response to antipsychotics was defined as a 2-point decrease in the CGI severity scale from baseline to follow-up. In general, women responded better than men, who needed higher doses than women. In this study, the widest gender gap was found in patients treated with 1st generation antipsychotics and clozapine. There was no gender difference in patients treated with risperidone. This is interesting in light of the fact that risperidone is associated with hyperprolactinemia, which reduces oestrogen levels, but it does not directly compare antipsychotic efficacy or tolerability pre- and post- menopause.

The highest level of evidence available as to choice of drug is provided by Goldstein and collaborators [16] who reanalysed an international randomized clinical 6-week trial (RCT) comparing the efficacy of olanzapine (weight gaining) and haloperidol (prolactin raising) to test gender differences in schizophrenia treatment response [14]. Patients were enrolled if they met the Diagnostic and Statistical Manual of Mental Disorders, Third Edition (DSM-III-R) criteria for schizophrenia related disorders. Exclusion criteria included the presence of other endocrine and central nervous system diseases. The initial sample consisted of 1337 patients treated with olanzapine and 658 treated with haloperidol. Both groups were comparable on age, sex, ethnicity, diagnostic subtype, and duration of illness. Psychopathological symptoms were assessed by using the Brief Psychiatric Rating Scale (BPRS), which evaluated specific positive, negative, affective, and anxiety symptoms. The study controlled for illness chronicity and menopausal status as both variables are capable of influencing antipsychotic response. The main findings were that premenopausal women showed significantly better antipsychotic response than women who were postmenopausal, irrespective of the drug they received and the chronicity of their symptoms.

The question about whether the deliberate selection of antipsychotics (prolactin raising vs. prolactin sparing; weight gaining, or weight sparing) in postmenopausal women with schizophrenia makes a difference to outcome remains essentially unanswered, although the analysis by Goldstein et al. [16] suggests that there may be no difference.

### 3.2. Antipsychotic Dose Adjustment in Postmenopausal Women with Schizophrenia and Related Disorders

RCTs assessing dose adjustments in postmenopausal women with schizophrenia compared to men and premenopausal women are lacking. Current theories rest on data of observational and cross-sectional studies.

Seeman [17] carried out a three-year survey (1978–1981) of patients who fulfilled diagnostic criteria for schizophrenia at the time and who were treated by the same psychiatrist over a 3-year period (1978–1981) [15]. In this study, the author accurately described that patients were only included if they met research diagnostic criteria (RDC) for schizophrenia. Antipsychotic medications were described as neuroleptics, in an attempt to explain the effects of oestrogens to potentiate neurolepsis in animal models or the antidopaminergic effects in humans. The sample consisted of 43 women and 58 men, and all patients had a history of at least two hospitalizations. The hypothesis was that women required lower doses of antipsychotics than men. Antipsychotic doses were converted to chlorpromazine equivalents per day. No statistically significant differences in antipsychotic doses were found between men and women. However, when stratified for age, younger women (under 40) needed lower doses than age-matched men, and women over age 40 needed higher doses. Assessment of need was made clinically. No rating scales were used. This was one of the first reports to hypothesize that a hormonal effect (the advent of menopause) could partially explain sex/age dose differences. 

Based on the ensuing oestrogen protection hypothesis, Shlomi Polachek and co-workers [18] compared the frequency of voluntary and involuntary hospitalizations, lengths of stay, and prescribed antipsychotic medication dose at discharge in men and women in two age groups: (a) between 18 and 45 years and (b) older than 55 years. The authors aimed to explore the relationship between sex and hospitalization in patients with psychotic disorders, not restricting the sample to those suffering from schizophrenia. The sample was divided into two groups according the treatment they were given: intramuscular risperidone or clozapine. The investigators hypothesized that women before menopause (<45 years) would have fewer involuntary hospitalizations than men, that their lengths of stay at hospital would be shorter, and that their medication dose would be lower than men’s. Some of the hypotheses proved to be correct. In the younger group, women did have fewer involuntary hospitalizations than men. In the older group, the frequency of involuntary hospitalizations was the same in men and women, but women were prescribed significantly higher mean intramuscular risperidone doses and non-significantly higher clozapine doses.

With gender-related theories in mind, González-Rodríguez and colleagues [19] explored potential clinical and reproductive variables capable of influencing antipsychotic response in a sample of postmenopausal women with schizophrenia. The investigators hypothesized that response to antipsychotics would change after menopause. The sample included 60 patients with schizophrenia and 4 with schizoaffective disorder, all fulfilling DSM-IV-TR criteria for both diagnoses, who were followed for 12 weeks in a prospective observational study, controlling for duration of reproductive years and time since menopause. Time since menopause was significantly negatively associated with overall antipsychotic response, suggesting a decline in antipsychotic response with age. A potential explanation was that oestrogens exert a neuroprotective effect during reproductive years that lasts for a certain time after menopause and then disappears. The same research team then tried to determine whether gonadal hormone levels, follicle-stimulating hormone (FSH), luteinizing hormone (LH) levels, and FSH/LH ratio correlated with clinical improvement in specific clinical symptoms [20]. They recruited 37 postmenopausal women with schizophrenia in a 12-week prospective observational outcome study, and were included if they met criteria according to the DSM-IV-TR. Neither this study nor the previous one specified which tests—laboratory or brain imaging tests—were underwent to exclude other organic psychoses. Partial correlational analyses revealed no statistically significant correlations between changes in psychopathological symptoms and hormone levels. For future investigations, the authors recommended considering the inclusion of adrenal hormones and analysing hormonal ratios rather than single hormone levels.

Although the abovementioned papers were cross-sectional or observational studies, and not RCTs, the overall findings appear to support the need of antipsychotic dose adjustment at the time of menopause.

### 3.3. Hormone Replacement Therapy in Postmenopausal Women with Schizophrenia

Hormone replacement therapies (HRT) replace ovarian hormones lost at menopause and they ameliorate symptoms attributable to menopause such as hot flashes, mood disturbance, sleep disturbance, and vaginal dryness. These potential effects may apply for many women, whether or not they suffer from schizophrenia, or whether they are diagnosed with other psychotic disorders. HRT is mainly considered safe for the first 5 years of menopause or longer, and the current evidence is that it will improve psychotic symptoms in at least some women [2]. The question about differences in response between early and late-onset women is not still answered.

As oestrogens exert many neuroprotective effects, they have been considered as potential adjunctive treatments for schizophrenia. However, RCTs evaluating the effect of adjunctive estradiol on women with schizophrenia have only been conducted in premenopausal women [21]. In general, the use of oestrogens in women poses risks for breast and uterine malignancy, although it has been suggested that it could diminish cardiovascular risks as well as dementia risks.

Hulley and collaborators conducted a randomized, blinded, placebo-controlled prevention trial as part of the Heart and Oestrogen/progestin Replacement Study (HERS) in postmenopausal women from outpatient and community settings testing cardiovascular effects of oestrogen plus progestin in women with pre-established coronary disease [22]. A total of 1380 women received hormones and 1383 received placebo. Treatment with oral oestrogen plus progestin was not associated with a reduction in coronary heart disease (CHD) in women with existing heart disease, suggesting that this treatment could not be recommended for secondary prevention of CHD. Thromboembolic events were not found to be higher.

The oestrogen plus progestin component of a randomized controlled primary prevention trial investigated major health benefits and risks [23]. The primary outcome was coronary heart disease, and the primary adverse outcome was invasive breast cancer. The authors found that health risks from the use of combined oestrogen plus progestin were superior to benefits in postmenopausal women, suggesting that this combination of treatment should not be started for primary prevention of cardiovascular heart disease (CHD).

Chlebowski and collaborators reported data on breast cancer risk assessment in 16,608 postmenopausal women aged 50–79 years without uterine illness [24]. Screening mammography and breast cancer clinical examinations were conducted at baseline and yearly, with a mean follow-up of 5.2 years (SD: 1.3). The group receiving oestrogen plus progestin had a higher incidence of breast cancer, and the cancer was found at a more advanced stage when compared with the placebo group.

Oestrogen replacement therapy is, therefore, not at present recommended for postmenopausal women with schizophrenia, especially as the incidence of breast cancer is significantly higher in schizophrenia than in the general population [25]. This systematic review and meta-analysis included cohort studies (prospective or retrospective, adult women with a diagnosis of schizophrenia. Diagnostic criteria were not specified in the inclusion and exclusion criteria section. Randomized controlled trials of oestrogen replacement in women with schizophrenia are still lacking.

### 3.4. Selective Estrogen Receptor Modulators for the Treatment of Postmenopausal Women with Schizophrenia and Related Disorders

Several studies have tested the combination of conjugated oestrogens with antipsychotics in premenopausal women. In a sample of 32 women with chronic schizophrenia, Ghafari and collaborators [26] evaluated the oestrogen’s effect in women with chronic schizophrenia who were hospitalized. Oestrogens were added as an adjunctive treatment. The sample was not divided according to the antipsychotic treatment they received.

The use of other estrogenic compounds has also been investigated as adjunctive treatment [27], one of the most prevalent ones being raloxifene, a selective oestrogen receptor modulator [28,29,30,31,32,33,34,35]. This drug has been tested for positive symptoms, negative symptoms, and cognitive symptoms in RCTs. The vast majority of them have found raloxifene to be safer compared to pure oestrogen compounds, and have found positive effects in treating psychotic symptoms.

Details of studies evaluating efficacy of adjunctive raloxifene in the treatment of postmenopausal schizophrenia women can be found in Table 1.

Kulkarni and collaborators [32] carried out a 12-week double blind randomized controlled trial of women with schizophrenia receiving raloxifene 120 mg/day (n = 13), raloxifene 60 mg/day (n = 9) or placebo (n = 13) [30]. Psychotic symptoms were assessed by the Positive and Negative Syndrome Scale (PANSS), at baseline and every two weeks (up to 12 weeks). Inclusion of patients was restricted to the DSM-IV criteria for schizophrenia, schizoaffective disorder, or schizophreniform disorder. The diagnosis was confirmed by the SCID interview. The status of menopause was confirmed by hormone levels and the presence of menopause symptoms using the Green Climacteric Scale. The group receiving raloxifene 120 mg/day showed higher levels of improvement in total and general psychopathological symptoms compared to the other two groups. This is in agreement with Usall and collaborators [34] who conducted a 12-week double-blind randomized placebo-controlled trial in 16 patients receiving raloxifene 60 mg/day and 17 treated with placebo [34]. Psychotic symptoms were measured by means of the PANSS scale at baseline and after 4 and 12 weeks. At endpoint, raloxifene 60 mg/day correlated with more improvement in positive, negative, and general psychopathological symptoms than placebo. An 8-week parallel-group placebo-controlled trial was conducted by Kiamehr and colleagues [31]. Women were enrolled if they had a diagnosis of schizophrenia according to the DSM-IV-TR criteria. The diagnosis was confirmed by the Structured Clinical Interview for DSM-IV-TR Axis I disorders (SCID), chart review and an interview by a senior psychiatrist. Patients receiving the combination of risperidone 6 mg/day and raloxifene 120 mg/day (n = 23) showed more improvement in positive psychotic symptoms compared to risperidone 6 mg/day plus placebo [31].

Huerta-Ramos et al. [30] from the Usall group carried out a 12-week double-blind randomized placebo-controlled trial with two comparison groups: raloxifene adjunct 60 mg/day and placebo adjunct [30]. Women were included in the clinical trial only if they met criteria for schizophrenia according to DSM-IV-TR classification. Forty-three per cent of the sample receiving raloxifene were also treated with second-generation antipsychotics. The authors also described that 7.7% received a combination of first and second-generation antipsychotics, in both groups: raloxifene and placebo groups. Psychopathological and cognitive symptoms were assessed by using the PANSS scale and a neuropsychological battery at baseline and at 12 weeks. Women treated with raloxifene 60 mg/day showed better improvement in memory and executive function than the placebo group. A 24-week double-blind randomized placebo-controlled trial confirmed the efficacy of adjunctive raloxifene 60 mg/day in negative and general psychopathological symptoms compared to postmenopausal women receiving placebo [35]. In a study of severely ill decompensated women with schizophrenia, a 16-week double-blind randomized placebo-controlled trial found that women receiving antipsychotics plus raloxifene 120 mg/day [35] showed no significant improvement in psychopathological symptoms compared to the placebo group [35]. Postmenopausal women fulfilled DSM-IV-TR criteria for schizophrenia or schizoaffective disorder and received, all of them, treatment with antipsychotics. A first group were treated with raloxifene, and the second received placebo. 

In 69 women with schizophrenia, 12 weeks of treatment with raloxifene 120 mg/day added to antipsychotics was found to be effective compared with placebo in terms of cognitive symptoms [29]. This study included results from two clinical trials recruiting women with DSM-IV criteria for schizophrenia and schizoaffective disorder. The efficacy of raloxifene was tested according to the menopause status and added to the use of antipsychotics. Samples were not grouped by antipsychotic type (typical or atypical). The authors stratified the sample by menopause status: pre-menopausal women, peri-menopausal women, and postmenopausal women. After adjusting for hormone levels, semantic fluency, picture naming, and list recognition were found to change significantly in patients receiving adjunctive raloxifene compared to the placebo group. More recently, Brand and collaborators [28] published a study protocol for the investigation of raloxifene augmentation in men and women with schizophrenia [28]. This randomized controlled trial included patients with a diagnosis of schizophrenia, schizophreniform disorder, schizoaffective disorder, or a psychotic disorder not otherwise specified, according to the DSM-IV criteria. Furthermore, the Mini International Neuropsychiatric Interview 5.0.0 was administered to confirm the diagnosis of psychotic disorder. The authors aimed to replicate earlier findings about the efficacy of adjunctive raloxifene in postmenopausal women. The study was designed to extend these findings to a population of men and pre- and perimenopausal women with schizophrenia who receive raloxifene 120 mg/day or placebo during 12 weeks. Results from the trial are still warranted. 

Two recent meta-analyses of RCTs [36,37] concluded that raloxifene may be a useful add on treatment for postmenopausal women with schizophrenia as long as symptoms are not too severe. Although raloxifene does have potential side effects, the trials have not found side effect differences between raloxifene and placebo adjunctive therapy. The meta-analysis carried out by Zhu et al. [36] including randomized, double blind controlled trials recruited participants diagnosed with schizophrenia by any diagnostic criteria. All patients received adjunctive raloxifene plus antipsychotics as an intervention and were compared with a group treated with antipsychotics plus placebo. Patients were not grouped according to the antipsychotic medication they received. The work by Wang et al. [37] included randomized clinical trials, not open label studies, enrolling women with schizophrenia according to DSM-IV.

The brain pathways through which oestrogen or its analogues improve antipsychotic response have yet to be discovered. One recent clue is that specificity protein 4 (SP4) transcription factors may be involved in raloxifene symptom improvement, suggesting that this protein could be useful as a potential biomarker of treatment response [38]. Labad and collaborators [33] demonstrated the relevance of genetic variants in UGT1A8 and oestrogen receptor 1 genes in the modulation of response to raloxifene in postmenopausal women with schizophrenia [33].

In summary, evidence from RCTs points to the use of adjunctive raloxifene as an effective and safe way to treat psychotic symptoms in postmenopausal women with schizophrenia. There are no comparisons, however, between pre- and post-menopausal women on the effect of added raloxifene on the requisite dose of antipsychotics, nor on a potential effect on antipsychotic adverse effects.

### 3.5. Influence of Menopausal Symptoms and Comorbidities on Treatment Outcomes in Schizophrenia and Related Disorders

The menopausal transition has been associated with the emergence of psychiatric symptoms in at least 20% of all women [39]. Several studies have categorized this period as one of high vulnerability for mood instability that can trigger the onset of bipolar disorder, or worsen affective symptoms in women with a pre-existing history of bipolar disorder. Women suffering from bipolar disorder with psychotic features are especially vulnerable in terms of psychopathological worsening in the menopausal transition, and they frequently need combined treatment with antipsychotics, in the course or maintenance of a manic phase or as a consequence of depressive episodes. However, few RCTs have investigated the impact of perimenopause, menopausal symptoms, or accompanying medical comorbidities on women with schizophrenia.

Gurvich and collaborators [29] addressed the influence of hormones of the hypothalamic-pituitary-gonadal (HPG) axis on cognitive performance in women with schizophrenia [29]. The sample consisted of two hundred and forty women with a diagnosis of schizophrenia or schizoaffective disorder from three separate clinical trials. Hormones included sex steroids and pituitary hormones: estradiol, progesterone, luteinising hormone, and follicle-stimulating hormone. Women were classified into four categories according to menopausal stage (perimenopausal, postmenopausal, premenopausal with regular cycles, and premenopausal with irregular cycles). Menstrual cycle irregularity was associated with poorer cognitive performance in verbal memory and fluency, and psychomotor speed. Perimenopausal women did not show cognitive changes, but postmenopausal women exhibited significantly poorer visuospatial performance than the other groups.

A recent nationwide population-based retrospective cohort study investigated the association between symptomatic menopausal transition and newly diagnosed psychiatric disorders, including schizophrenia, affective disorders (bipolar and depressive disorders), anxiety, and sleep disorders [40], as compared to a cohort without menopausal symptoms. The first group showed a significantly higher incidence of newly diagnosed bipolar and depressive disorders, anxiety, and sleep disorders, but rates of schizophrenia were not affected. RCTs have not confirmed these interesting findings.

With regard to medical comorbidities at the time of menopause, when oestrogen levels fall, the risk of venous thromboembolism (VTE) increases. With this finding in mind, a recent population-based nested case–control study investigated whether the use of antipsychotics was associated with an increased VTE risk [41]. The authors did not restrict their sample to a group of patients with schizophrenia fulfilling ICD or DSM criteria. The use of antipsychotics was, indeed, significantly positively associated with VTE in a dose dependent manner. Obstructive sleep apnoea (OSA) is also a relevant comorbid condition in menopausal women with schizophrenia. Seeman [42] highlighted the importance of an early diagnosis and adequate treatment of obstructive sleep apnoea (OSA) in these women. 

After menopause, women suffer changes in bone mineral density which can be increased by other concurrent factors. The use of psychotropic medications appears to increase this risk [43]. This was not found by Renn et al. [44] to be associated with menopause. Their assumption was that BMI was already low in women with schizophrenia prior to menopause, presumably due to the hyperprolactinemia induced by antipsychotics. In the study of Renn [44], all patients met DSM-IV criteria for schizophrenia, and those suffering from severe extrapyramidal symptoms or poor disease control, or those having medical or mental disability were excluded from the study. Aripiprazole, which is known to be a prolactin-sparing antipsychotic, has been advocated in this group of women [45] but there are, to date, no RCTs to support this recommendation. 

Menopause knowledge, expectations of the effect of menopause, and menopause-related quality of life have been investigated in postmenopausal women with schizophrenia (n = 30), bipolar disorder (n = 25), and major depression (n = 36) [46]. The most frequent symptoms experienced at menopause were found to be depression, anxiety, fatigue, anergia, and poor memory [46]. As such symptoms influence mental health, the recommendation is to assess them carefully and treat them appropriately in women with severe mental illness [47]. 

The association between cancer, mental illness, and mortality in this age group of women has also been investigated. Theories have been constructed on the basis of observational or cross-sectional studies. In a sample of 113 patients with gynaecological cancer, Lau and co-workers [48] found that 31% suffered from depressive disorders, 16% from anxiety disorders, and 2% from schizophrenia. All women fulfilled DSM-IV criteria for Axis I Disorders according to the Structured Clinical Interview for DSM-IV, Patient Research Question. Importantly, women with schizophrenia and schizoaffective disorder, without a substance use disorder or cognitive impairment, receive less frequent mammograms, pelvic examinations, and Pap tests compared to women without a psychiatric diagnosis [49]. More recently, Lawrence et al. [50] found that, post-menopause, women with schizophrenia show very high rates of breast cancer mortality. There are likely to be many reasons for this, but one potential reason, not investigated to date, is the negative impact of a cancer diagnosis on vulnerable women such as women with schizophrenia at the time of menopause.

Neurological disorders can also first emerge, or at least worsen, at menopause in women with schizophrenia. Tardive dyskinesia is a good example. Women with late onset schizophrenia (LOS) and men with early onset schizophrenia (EOS) showed more severe dyskinesia than women with EOS and men with LOS, suggesting that hormonal status may contribute to the severity of this movement disorder in menopausal women through an effect of oestrogen loss on striatal dopamine receptors [51,52]. 

In summary, during perimenopause, symptoms associated with the menopausal transition may worsen psychiatric symptoms in women with schizophrenia. There are no studies, however, that show that attending to menopausal symptoms improves outcomes. 

### 3.6. Family Stress, Psychopathological Symptoms, and Clinical Outcomes in Postmenopausal Women with Schizophrenia

At the time of menopause, women with schizophrenia suffer from changes at the family and social support level, which can potentially influence treatment outcomes. 

A recent study by Li and collaborators [53] investigated the current status of social functioning in a community sample of patients with schizophrenia. The influence of marital status on social dysfunction was particularly analysed. The authors found that never-married patients showed higher social dysfunction compared to those married in terms of self-care and occupational roles, suggesting that family support is crucial for clinical outcomes [53]. Another stressful situation is the increased risk of miscarriage that women with psychiatric disorders have. A recent work examined the risk of miscarriage in women with 12 different psychiatric diagnoses, and concluded that these populations show a higher risk of miscarriage that needs special care and service planning [54].

On the other hand, menopause is a significant event in the life of women. A survey of 39 patients with mental illness, 20% of whom were women with schizophrenia, explored the perceived effects of menopause on illness course from the patient and family perspective [55]. Patients with schizophrenia and their family members saw menopause as less significant than did patients with other psychiatric disorders and their families. This may be as treatment with antipsychotics induces amenorrhea early, and makes the permanent stopping of menses less memorable. However, it may also indicate a measure of distance between patients with schizophrenia and their family members. Physical examination or neurological investigation in included women were not detailed in this study.

Caring for family members with schizophrenia is stressful for family caregivers and sometimes difficult for both parts. Expressed emotion and hypercriticism refer to caregiver’s attitude towards an individual suffering from schizophrenia and have been hypothesized to be psychosocial stressors potentially associated with recurrence of illness and psychotic exacerbations [55,56]. Expressed emotions include criticism, hostility, and emotional over involvement. Good support from health and social services may help family caregivers overcome stress, as shown by a recent cross-sectional study [56]. 

No RCTs have explored the association between family stress and clinical outcomes in postmenopausal women with schizophrenia.

In summary, menopausal women with schizophrenia need extra help with responding to health needs, as well as to all other needs they may have at this age (housing, marital relationships, financial issues, or problems with children) so that social support from relatives becomes very important, perhaps more important with respect to worsening symptoms than hormones. The conjunction of the two probably explains the increase in psychotic symptoms at this time. Recent reports emphasize that age-related psychosocial factors may not only worsen symptoms, but may also trigger new onset of schizophrenia at this stage of life [56].

### 3.7. Specialized Care Settings for Peri- and Post-Menopausal Women with Schizophrenia

Very few studies have explored the efficacy of specific settings in meeting the many needs—pharmacologic, occupational, psychologic, family, social, and general health—of women with severe mental illness who are undergoing or have undergone menopause.

Following menopause, women may experience new health problems. This has already been addressed earlier. In the case of osteopenia and osteoporosis, for example, as in several other medical conditions, screening, preventive treatment, diagnosis, and prompt medical or surgical intervention becomes necessary [57]. Mental healthcare settings affiliated with specialty health clinics become vital [58]. Prevention of obesity is another vital concern as both menopause and antipsychotic treatment induce weight gain. Care settings need to be able to provide expert drug treatment, nutritional guidance, routine laboratory testing for cholesterol levels and diabetes markers, routine blood pressure measurements, exercise programs, routine weight and waist measurement, and health advocacy. Some programs on large grounds even promote patients growing their own vegetables as obesity is dangerously associated with metabolic and cardiovascular risk (diabetes mellitus, hypertension, and coronary heart disease). Skouroliakou et al. [59] investigated the effect of a nutritional intervention on body weight as well as on blood levels of glucose and lipids in 25 obese postmenopausal women treated with antipsychotics and compared their results with those of 28 obese but mentally well women. Women with schizophrenia were diagnosed according to the DSM-IV criteria, and the diagnosis was confirmed by a senior psychiatrist. The absence of any chronic disease and alcohol use were inclusion criteria. Outcome measures were assessed at baseline and after three months of treatment. Patients unfortunately showed lower reduction in waist circumference and glucose and lipid levels compared to the control group [59].

Substance use disorders, while always a problem in the schizophrenia population, become more prevalent at menopause due to the many difficulties women experience at this time. Family and childcare are further necessities [46]. There are often added dental needs, housing needs, economic needs, and occupational needs at this time. The setting of care needs to be able to provide effective multidisciplinary, comprehensive treatment. 

What becomes especially relevant to recovery, defined as an individual’s ability to cope and adapt to daily life events and stressors, is social skills training. A cross-sectional, descriptive correlational study in 129 patients with severe mental illness investigated the impact of such training on psychiatric symptoms and clinical outcomes. The study concluded that psychoeducation programs to teach life skills provided by clinical nursing staff to postmenopausal women was effective [60]. It can sometimes be provided remotely to patients’ homes.

Considering ideal treatment settings, home care programs have been suggested for older women with acceptable functionality and intact cognition [61]. In older age, patients prefer to stay at home and to avoid hospitalization. As long as medical or substance use problems or aggressive outbursts do not need attention, home visiting by a multidisciplinary mental health team has been recommended as an excellent option [62].

Family and friend support is very important for the management of any disease. There are many homeless women with schizophrenia of menopausal age who lack social support and are best treated in settings where they can meet others and, ideally, learn to form social networks [63]. Partial hospitalization programs may be ideal [64].

In conclusion, interventions for women with schizophrenia differ by stage of life. At all stages, preventive strategies must be prepared to address the risks and benefits of treatment, lifestyle modifications, prevention of medical comorbidities, life skill training, and mobilization of psychosocial support.

## 4. Discussion

As oestrogen loss at menopause profoundly affects the expression of psychosis and its response to drug treatment, this review is based on the premise that women with schizophrenia in peri- and post-menopause need specialized treatment. For this reason, we searched the current literature to find answers to the following questions: Does the choice of antipsychotic (prolactin sparing vs. prolactin raising; weight gaining vs. weight neutral) make a difference? Does raising the dose over premenopausal levels make a difference? Does adding hormone replacement therapy or selective oestrogen receptor modulator make a difference? Does attending to comorbidities or family stress make a difference? Does family stress increase symptom level? Does treatment setting and the comprehensiveness of care make a difference?

Schizophrenia is a serious mental illness that affects around 1% of the world population [65]. Later in life, schizophrenia has been increasingly diagnosed, particularly in women. A high amount of literature has pointed out that aging effects and hormonal changes that occur at the menopausal period may have influenced the prevalence of later diagnoses [65]. As people age, the incidence of other psychotic disorders also increase [2,3]. This is the particular case for patients with delusional disorder who show an onset of disease at the 40–45 years [3].

In our review, we found that the vast majority of studies included mixed samples formed by patients with schizophrenia and schizoaffective disorders [19,20,21,29,35,49], very few studies are restricted to patients with schizophrenia [15,16,17,25,30,31,33,36,37,44], some others include a wide range of psychotic disorders [18,28,32,47], and other studies include patients with severe mental illness without specific details of the diagnosis of schizophrenia related disorder. Prevalence of each category of diagnosis is difficult to calculate, as several studies grouped patients of different diagnosis. For instance, this is the case of the research combining schizophrenia and schizoaffective disorder patients.

Overall, we found that the vast majority of evidence supports the suggestion that women need higher antipsychotic doses after menopause [19], although a recent review found few studies that address this specific point [66]. Lange and collaborators [67] have recommended dose adjustments during the menstrual cycle in premenopausal women, strongly suggesting that hormone alterations affect antipsychotic response. They also suggested that first-generation antipsychotics should not be recommended as a first line treatment for postmenopausal women as they prolong QTc interval and increase prolactin levels. The caveat, however, applies to many antipsychotics as most increase the QTc interval and risperidone, a 2nd generation drug, is associated with hyperprolactinemia. Nevertheless, the lesson to clinicians is to check frequent side-effects carefully before prescribing medications to postmenopausal women, and to consider switching to new medications at this stage of women’s lives. There is, however, a lack of RCTs that have investigated whether the choice of antipsychotic and adjustment of dose in postmenopausal women with schizophrenia is relevant to outcome.

The third and the fourth question were focused on the use of hormone replacement therapy, therapeutic oestrogen or raloxifene in addition to antipsychotics in postmenopausal women with schizophrenia. As oestrogen compounds present safety problems, raloxifene has been more widely investigated [36]. Evidence from RCTs confirms that raloxifene at the doses used was both safe and effective for some of the symptoms of schizophrenia. A recent meta-analysis of five RCTs confirmed that the raloxifene group and the placebo group showed similar rates of adverse events and discontinuation [36]. In terms of efficacy, this work highlighted that women receiving raloxifene may improve in positive, negative, and general psychopathological symptoms compared to the placebo group. The vast majority of studies include women with schizophrenia, but others are formed by a combination of schizophrenia and schizoaffective women. 

The question about the effect of menopausal symptoms and comorbidity in menopausal women with schizophrenia leads to the conclusion that sleep apnoea, osteoporosis, and cardiovascular problems, as well as substance use disorders, are all common comorbid burdens for women at this stage of life [36,40,43,47,68,69]. RCTs evaluating the effect of such comorbidities on clinical outcomes in postmenopausal schizophrenia are, however, lacking. The presence of comorbid physical illnesses and substance use disorders is relevant, as they have an effect on adherence with medications and other clinical outcomes.

The risk of cancer also increases at this age but, despite higher rates of obesity, smoking, and substance abuse, mid age cancer is probably not more prevalent in patients with schizophrenia than it is in the general population, although the mortality rate is higher [70]. This has been attributed to patient factors (negative symptoms, cognitive factors, and social isolation), provider factors (diagnostic overshadowing, exclusion from trial drugs) and health system factors (medical/psychiatric silos). For instance, Lindamer and collaborators [51] found that women with schizophrenia received fewer gynaecological services compared to other women, suggesting that interventions should be provided to address this disparity in screening services.

The frequency of central nervous system disorders such as cognitive disturbances or tardive dyskinesia (TD) are increased at the time of menopause parallel to oestrogen decline [51,52]. There are now treatments for TD, which may explain why oestrogen supplementation has not been tested for this condition, but it has been tested for cognitive decline. Raloxifene has been shown in some studies to improve cognitive deficits in postmenopausal women with schizophrenia [28,29,30,35].

The sixth question we addressed was the role of family and other social supports on patient stress levels and, by implication, on symptom levels. Caregiver stress levels have also been shown to influence patients’ symptom scores. Support from health and social professionals is recommended, despite the fact that no RCTs have evaluated the effect of specific interventions to ameliorate stress in family members in this population [54]. Marital status is found to be a relevant mediator of social functioning in women with schizophrenia and related disorders, and miscarriage (another psychosocial risk factor) appears to be higher in these populations [53,54]. By implication, in several cases, it can be assumed that women with schizophrenia present lower support and higher psychosocial risk factors, which have an influence on clinical outcomes.

In terms of ideal treatment settings, there are advantages to home care, to partial hospitalization programs within a general hospital framework, and to community settings. More important than the setting itself is the multidisciplinary team and the potential of comprehensive treatment which might include psychological and pharmacologic intervention, medical specialist availability, psychoeducation, cancer, osteoporosis and sleep apnoea screening, social skills training, family care, exercise programs, nutritional guidance, housing referral, smoking cessation programs, weight watching, substance withdrawal [71], and eHealth group counselling or coaching programs [72]. 

### Strengths and Limitations of the Review

To the best of our knowledge, this is the first review to specifically investigate whether meeting the specific needs of menopausal and postmenopausal women improves their outcomes, both with respect to cognition and positive symptoms of psychosis. We addressed seven issues whose resolution could potentially improve outcomes: choice of antipsychotic, need for dose adjustment, hormone use, treating menopausal symptoms, screening for, diagnosing, and treating medical comorbidities, attending to the patient’s social support system, and selecting the appropriate treatment setting.

The main limitation is the scarcity of RCTs that address our seven issues. The highest level of evidence is for the use of adjunctive raloxifene.

Future studies should be focused on the investigation of clinical outcomes in specific samples formed by patients with schizophrenia and other related disorders. The vast majority of studies include mixed samples of women with schizophrenia and schizoaffective disorders, or even they include other psychotic disorders such as delusional disorder. Furthermore, few information is available with regard to the prevalence of schizophrenia and other related disorders at this period of women’s life. At the time of menopause, a second peak of incidence has been observed in women with schizophrenia, and delusional disorder appears later in life.

## 5. Conclusions

The selection of antipsychotics (prolactin raising vs. prolactin sparing; weight gaining vs. weight sparing) for the treatment of postmenopausal women with schizophrenia and related disorders is one of the major challenges in gender approaches to treatment and remains unresolved due to the lack of randomized-controlled trials. The need for dose increases at the time of menopause comes from observational and cross-sectional studies. Current evidence supports the fact that oestrogens exert beneficial effects on mental health. The use of HRT, however, is controversial, due to tolerability and safety concerns. The addition of raloxifene, a selective oestrogen receptor modulator (SERM), to a standard antipsychotic regimen appears to improve outcomes, but the crucial question of whether its use can reduce antipsychotic dose and, thereby, reduce the severity of adverse effects of antipsychotics, has not been tested. 

Medical comorbidities, such as venous thromboembolism, osteoporosis and sleep apnoea, and obesity, increase at the time of menopause, and antipsychotics increase the risks. The incidence of neurological disorders and cancers at this time undoubtedly influence clinical outcomes in schizophrenia, but this has not been confirmed by RCTs. In spite of the overall lack of a high level of evidence, we recommend intervention at the level of family, prevention and vigorous treatment of medical comorbidities, the encouragement of healthy lifestyle modifications and the provision of psychosocial support in care settings appropriate to this population.

## Figures and Tables

**Figure 1 jpm-11-00849-f001:**
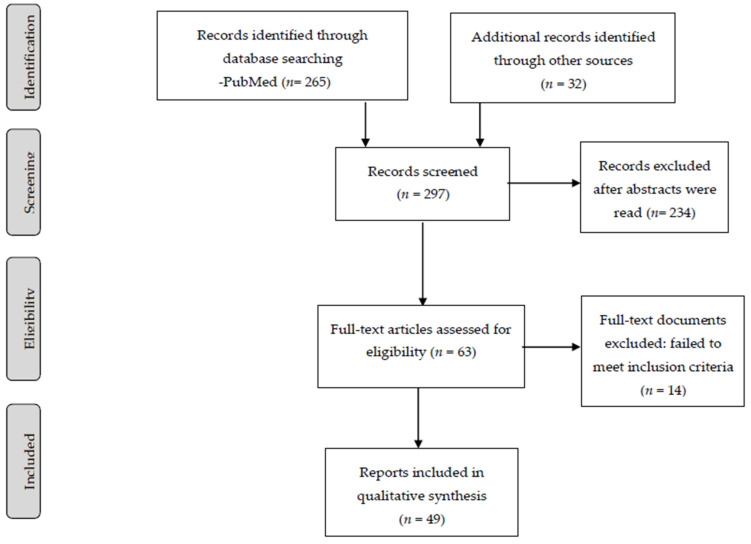
Flow diagram of included reports.

**Table 1 jpm-11-00849-t001:** Most relevant studies investigating the efficacy of raloxifene in the treatment of postmenopausal schizophrenia women (*n* = 8).

Authors and Year of Publication	Study Design	Comparison Groups	Psychometric Instruments	Findings
Kulkarni et al., 2010 [32]	12-week double-blind randomized controlled trial	Raloxifene 120 mg/day (*n* = 13) vs. raloxifene 60 mg/day *(n* = 9) vs. placebo (*n* = 13)	Psychopathology: PANSS and MADRS (at baseline, weeks 2, 4, 6, 8, 10, and 12)	Raloxifene 120 mg/day was associated with improvement in PANSS total and general symptoms compared to raloxifene 60 mg/day and placebo
Usall et al., 2011 [34]	12-week double-blind randomized placebo-controlled trial	Raloxifene 60 mg/day (*n* = 16) vs. placebo (*n* = 17)	Psychopathology: PANSS (at baseline, weeks 4 and 12)	Raloxifene 60 mg/day associated with improvement in positive, negative, and general psychotic symptoms
Kianimehr et al., 2014 [31]	8-week parallel-group placebo-controlled trial	Risperidone 6 mg/day plus raloxifene 120 mg/day (*n* = 23) vs. risperidone 6 mg/day plus placebo (*n* = 23)	Psychopathology: PANSS (at baseline, weeks 2, 4, and 8)	Patients receiving the combination of risperidone and raloxifene 120 mg/day presented higher improvement in positive symptoms than placebo
Huerta-Ramos et al., 2014 [30]	12-week double-blind randomized placebo-controlled trial	Raloxifene 60 mg/day (*n* = 16) vs. placebo (*n* = 17)	Psychopathology: PANSS (at baseline, week 12) Cognition: Neuropsychological battery (at baseline, week 12)	Women receiving adjunctive raloxifene 60 mg/day presented improvement in memory and executive function
Labad et al., 2016 [33]	24-week double-blind randomized parallel placebo-controlled trial	Raloxifene 60 mg/day (*n* = 36) vs. placebo	Psychopathology: PANSS (baseline, weeks 4, 12, and 24) Genetics: ESR1 and UGT1A8 gene	Genetic variants in UGT1A8 and ESR1 genes modulated, respectively, response to raloxifene in terms of negative and general symptoms
Weiser et al., 2017 [35]	16-week double-blind placebo-controlled RCT	Raloxifene 120 mg/day vs. placebo	Psychopathology: PANSS, CGI Cognition: Composite Brief Assessment of cognition	Severely ill decompensated patients receiving antipsychotics plus raloxifene showed no significant improvement in psychopathology
Gurvich et al., 2019 [29]	Pooled data from two 12-weeks RCT	Raloxifene 120 mg/day (*n* = 33) vs. placebo (*n* = 36)	Psychopathology: PANSS, MADRS Cognition: RBANS	When accounting for menopause status and hormone levels, women treated with raloxifene presented higher changes in cognitive performance compared to placebo
Brand et al., 2020 [28]	12-week double-blind placebo-controlled, double-blind, RCT	Raloxifene 120 mg/day vs. placebo	Psychopathology: PANSS Cognition: BACS	Study protocol (not finished): including men and pre/perimenopausal women

Abbreviations: BACS, Brief Assessment of Cognition in Schizophrenia; CGI, Clinical Global Impression Scale; ESR1, Oestrogen Receptor 1; MADRS, Montgomery–Asberg Depression Rating Scale; PANSS, Positive and Negative Syndrome Scale; RBANS, Repeatable Battery for the Assessment of Neuropsychological Status; RCT, Randomized Clinical Trial; SANSS, Scale for the Assessment of Negative Symptoms; and UGT1A8, UDP-glucuronosyltransferase 1A8.

## Data Availability

The data presented in this review are available on request from the corresponding author.

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
