# Peer review of "Peri- and Post-Menopausal Women with Schizophrenia and Related Disorders Are a Population with Specific Needs: A Narrative Review of Current Theories"

_jpm, 2021, doi:10.3390/jpm11090849_

Round 1

Reviewer 1 Report

I would like to thank the authors for sharing the results of their efforts. The submitted draft was adequately focused on its main subject, it offered a balanced analysis of the issue and within the scope of the selected format. I have summarized my main concerns in the following bulletin points: 1)Row 98 “[…]This From the initial 265[…]” the sentence should be edited as it looks like the pronoun “This” was inserted by mistake; 2) Table 1 appears to be truncated, as I cannot manage to display the last cell.

Author Response

We would like to thank the Reviewer 1 for the comments and suggestions that have improved the quality of the manuscript. We marked in yellow in the body of the manuscript, all changes me made, and listed them in the following lines.

“I have summarized my main concerns in the following bulletin points”

1) Row 98 “[…] This From the initial 265[…]” the sentence should be edited as it looks like the pronoun “This” was inserted by mistake.

We have corrected this mistake by removing out “This”.

2) Table 1 appears to be truncated, as I cannot manage to display the last cell.

We have modified Table 1 with the aim of reducing irrelevant information and the Table’s size.

Reviewer 2 Report

Affiliations: I would recommend not to translate the departments designations. I would also suggest to explain the acronyms, eg UAB, CIBERSAM.

Abstract: In the Methods section the authors should include the whole time length of the PubMed search, eg 1900-2021? Results are almost telegraphic, eg it would be hard to Readers to understand why HRT are not recommended in postmenopausal women: why? do you mean in all women? Or only in the women with schizophrenia? And what about women with other kind of psychosis...

Introduction: Language style should be less literary, eg what the Authors mean by “to keep psychotic symptoms at bay”? Another example would be “in ovariectomized rats, co-administration of 17 beta-estradiol with haloperidol and clozapine increases the positive effect”. What effect? The whole antipsychotic effect? Or only the effect on positive symptoms? Introduction is poor. It should include a proper historical perspective regarding menstrual psychoses, post-partum psychoses, post-menopausal psychoses and other hormone related psychoses in the female patients. 

Methods: The Authors should present a Figure explaining the Articles selection. I would recommend somekind of a  flowchart, explaining how many papers were included versus excluded along the way. On the other hand it would also be important to use PRISMA guidelines, in order to do a systematic review instead of a narrative one. We need more systematic reviews. There is no need for another narrative review regarding this topic, I am afraid.

Results: First sentence is hard to understand as it has two words starting with a capital letter “This” and “From”. Authors made a difference between first and second generation antipsychotics. And what about the third generation antipsychotics (aripiprazole, brexpripazole and cariprazine?). It would be also of uttermost importance to explain the readers the difference (if any at all!) between major tranquilizers, neuroleptics and antipsychotics. And, last but not least, acknowledge that different authors also make a distinction of typical versus atypical antipsychotics. Drug designations changed in the last decades and that is important in order to understand what kind of medication was used in all and every of the studies cited in the review.  Any review regarding the use of this kind of drugs should include the use of those different terms and concepts. History of psychopharmacology is always important.

Beware about the wrong use of acronyms, eg European Quality of Life Scale should be explained properly. The same happened with DSM-III that should be explained, and it was not. At line 194 the Authors stated “RCT’s are needed” but I believe that is not a result. This kind of commentaries would be more adequate for the conclusion section of the manuscript. Regarding “family stress” it would be important to add some commentaries about hypercriticism and high degrees of expressed emotion. A short discussion regarding marital status and the existence of miscarriage and/or childrens in the studied women should also be included in the text.

Table 1: I would recommend to avoid the use of vertical lines, in order to obtain more aesthetic elegance in Tables. Table is bigger than the size of the page so I was not able to see all columns. I would recommend to shorten text in every column in order to have a smaller Table 1. It should occupy no more than a single page. Please avoid having a Table divided by two different pages. This is not easy for a Reader to follow though. In the "Comparison Groups" column do not use indented text, nor a), b), c) bullets. I would recommend "vs." instead, separating the groups.

Discussion: discussion is poorly written. It should remind the readers that in the last century a lot has changed regarding the diagnosis of schizophrenia, in general, but also particularly among post-menopausal women. The authors should explain, for every mentioned study, which criteria for schizophrenia were used: ICD9, ICD10, ICD11 vs DSM-III, DSM-IV-TR or even DSM-5? The Authors should also explore the amount of studies that did a complete list of exams in order to exclude organic psychosis. Many of those studies include patients with other type of non-affectve psychosis, misdiagnosed as schizophrenia. I am afraid that many of those poor women should have been submitted to EEG to exclude temporal lobe epilepsy, MRI brain scan to exclude CNS tumor or even lumbar puncture to exclude encephalitis. Most of them where not studied in proper way and therefore I do not accept the diagnosis of schizophrenia among them. Other important problem is the contamination of those samples by other nosological entities, such as late onset schizophrenia (LOS), very late onset schizophrenia (VLOS),  paranoia, paraphrenia, delusional disorder or even schizoaffective disorder. Were these patients included in these studies? And what about the old sugtypes of schizophrenia, eg hebephrenic, paranoid, simple, undiferentiated, residual, etc... Would be better to accept this kind of limitations and to redo the review or maybe rename the article? Most of the included patients were not schizophrenic. They were psychotic, for sure, but not all were schizophrenic. And that is an enormous bias, dooming all efforts in schizophrenia research.  A review like this one is useless without presenting the prevalence of these syndromes among the studied patients, or at least  the Authors should express their opinion regarding this wicked problem. Authors should confront their epistemological perspective against the one mentioned (or not!) by the  Authors they cited in the Bibliography.

Author Response

We would like to thank the Reviewer 2 for the comments and suggestions that have improved the quality of the manuscript. We marked in yellow in the body of the manuscript, all changes me made, and listed them in the following lines.

  1. Affiliations: I would recommend not to translate the department designations. I would also suggest to explain the acronyms, eg UAB, CIBERSAM.

We have clarified our affiliations and explained the acronyms.

Abstract:

  1. In the Methods section the authors should include the whole time length of the PubMed search, eg 1900-2021?

We have included the time length applied for the electronic searches in the methods section and the abstract (1900-2021).

3.Results are almost telegraphic, eg it would be hard to Readers to understand why HRT are not recommended in postmenopausal women: why? do you mean in all women? Or only in the women with schizophrenia? And what about women with other kind of psychosis…

We entirely agree with reviewer 2 that evidence about the efficacy of hormone replacement therapy should be better clarified, and better explained in the section about hormone replacement therapy in postmenopausal women with schizophrenia and related disorders. We have added some paragraphs at the beginning of the 3.3. section.

“The potential effects of HRT may apply for many women, whether or not they suffer from schizophrenia, or whether they are diagnosed with another psychotic disorders. HRT is mainly considered safe for the first 5 years of menopause or longer, and the current evidence is that it will improve psychotic symptoms in at least for some women. The question about differences in response between early and late-onset women is not still answered.”

Introduction:

4.Language style should be less literary, eg what the Authors mean by “to keep psychotic symptoms at bay”? Another example would be “in ovariectomized rats, co-administration of 17 beta-estradiol with haloperidol and clozapine increases the positive effect”. What effect? The whole antipsychotic effect? Or only the effect on positive symptoms?

We agree with reviewer 2 that some aspects of language style should be modified. We have revised the whole manuscript, and specially modified these two examples.

5.Introduction is poor. It should include a proper historical perspective regarding menstrual psychoses, post-partum psychoses, post-menopausal psychoses and other hormone related psychoses in the female patients.

We agree with reviewer 2 that is necessary to introduce the menstrual psychoses and postpartum psychoses before describing in depth postpartum psychoses, as they all share a common biological pathway. We have added a paragraph introducing all of these entities with the aim of clarifying them and expand the introduction.

6.Methods: The Authors should present a Figure explaining the Articles selection. I would recommend some kind of a flowchart, explaining how many papers were included versus excluded along the way. On the other hand it would also be important to use PRISMA guidelines […].

Although this is a narrative review, we have applied key-terms, inclusion and exclusion criteria that have been further described in the methods section. We entirely agree with Reviewer 2 that we should describe with more details the screening and selection processes. We have added a Figure 1 (flow chart) and a better description of the included studies.

7.Results: First sentence is hard to understand as it has two words starting with a capital letter “This” and “From”.

We have deleted “This”. It was a mistake.

8.Authors made a difference between first and second generation antipsychotics. And what about the third generation antipsychotics (aripiprazole, brexpripazole and cariprazine?). It would be also of uttermost importance to explain the readers the difference (if any at all!) between major tranquilizers, neuroleptics and antipsychotics.

We have added two paragraphs at the beginning of the introduction section to clarify these concepts in a historical perspective.

9.And, last but not least, acknowledge that different authors also make a distinction of typical versus atypical antipsychotics. Drug designations changed in the last decades and that is important in order to understand what kind of medication was used in all and every of the studies cited in the review.  Any review regarding the use of this kind of drugs […]

At the beginning of the results section, we have clarified these concepts, and added a couple of references. We have also included the terminology that the studies used to designate drugs , in the body of the manuscript.

10.Beware about the wrong use of acronyms, eg European Quality of Life Scale should be explained properly. The same happened with DSM-III that should be explained, and it was not.

We have clarified it in the body of the manuscript at the first time where these acronyms appear.

11.At line 194 the Authors stated “RCT’s are needed” but I believe that is not a result. This kind of commentaries would be more adequate for the conclusion section of the manuscript.

We have removed this sentence “RCTs are needed” at the end of the section 3.3. and 3.5. We entirely agree with Reviewer 2 that this is not a result.

12.Regarding “family stress” it would be important to add some commentaries about hypercriticism and high degrees of expressed emotion.

We agree with reviewer 2 that some commentaries should be made to introduce criticism and expressed emotion in the process of caring for patients with schizophrenia.

13.A short discussion regarding marital status and the existence of miscarriage and/or childrens in the studied women should also be included in the text.

We have included a brief discussion about the role of marital status on psychosocial functioning in women with schizophrenia and related disorders. Furthermore, we discussed the existence of miscarriage in the studied population, as suggested by reviewer 2.

14.Table 1: I would recommend to avoid the use of vertical lines, in order to obtain more aesthetic elegance in Tables. Table is bigger than the size of the page so I was not able to see all columns. I would recommend to shorten text in every column in order to have a smaller Table 1. It should occupy no more than a single page. Please avoid having a Table divided by two different pages. This is not easy for a Reader to follow though. In the "Comparison Groups" column do not use indented text, nor a), b), c) bullets. I would recommend "vs." instead, separating the groups.

We agree with Reviewer 2 that Table 1 is too long and should be reduced in order to make it easier to understand for the readers. We have omitted some information and followed-up the suggestions.

Discussion:

15.The authors should explain, for every mentioned study, which criteria for schizophrenia were used: ICD9, ICD10, ICD11 vs DSM-III, DSM-IV-TR or even DSM-5?

We have explained in the results section, for very study, which diagnostic criteria were applied by the authors for schizophrenia patients and other related disorders. This point has been also included in the discussion section.

16.The Authors should also explore the amount of studies that did a complete list of exams in order to exclude organic psychosis. Many of those studies include patients with other type of non-affectve psychosis, misdiagnosed as schizophrenia. I am afraid that many of those poor women should have been submitted to EEG to exclude temporal lobe epilepsy, MRI brain scan to exclude CNS tumor or even lumbar puncture to exclude encephalitis. Most of them where not studied in proper way and therefore I do not accept the diagnosis of schizophrenia among them.

We have included information for each study about diagnostic criteria applied for the authors, and about which laboratory and other tests have been applied. We have also added this point as a limitation in the section of “Limitations and strengths”.

17.Other important problem is the contamination of those samples by other nosological entities, such as late onset schizophrenia (LOS), very late onset schizophrenia (VLOS),  paranoia, paraphrenia, delusional disorder or even schizoaffective disorder. Were these patients included in these studies? And what about the old subtypes of schizophrenia, eg hebephrenic, paranoid, simple, undiferentiated, residual, etc... Would be better to accept this kind of limitations and to redo the review or maybe rename the article?

We have described diagnostic criteria and which nosological entities have been considered in the included studies. This has been also added as a limitation of the article and we have renamed the article by focusing research on “Schizophrenia and Related Disorders”.

18.A review like this one is useless without presenting the prevalence of these syndromes among the studied patients, or at least the Authors should express their opinion regarding this wicked problem. Authors should confront their epistemological perspective against the one mentioned (or not!) by the Authors they cited in the Bibliography.

We have included a discussion about the prevalence of other nosological entities related to schizophrenia in peri- and postmenopausal women. We have also added some lines about our perspective and point of view.

Round 2

Reviewer 2 Report

Thank you for the Authors efforts following my suggestions.